# Tourist Perspectives on Agricultural Heritage Interpretation— A Case Study of the Qingtian Rice-Fish System

**Mingming Su [1], Menghan Wang [1], Yehong Sun [2,\*] and Ying Wang [2,3]**

1    School of Environment and Natural Resources, Renmin University of China, Beijing 100872, China
2    Tourism College, Beijing Union University, Beijing 100101, China
3    Rural Development Institute, Zhejiang Academy of Agricultural Sciences, Hangzhou 310021, China
*    Correspondence: sunyehong@buu.edu.cn

**Abstract:** The interpretation of an agricultural heritage system significantly affects the communication and connection between tourists and heritage sites. Taking the rice-fish system at Qingtian as an example, this study applies field investigations, a tourist questionnaire survey, and face-to-face in-depth interviews to explore agricultural heritage interpretation through the gaze of tourists. A two-dimensional framework integrating interpretation contents and forms for agricultural heritage systems is developed and adopted to guide the analysis. Research results show that tourists' overall recognition of agricultural heritage features of the Qingtian rice-fish system is not high. Regarding interpretation contents, the rice-fish agricultural landscape, traditional cuisine, and biodiversity exhibit higher awareness. Compared to the static and interactive interpretations, participatory interpretations demonstrated high effectiveness in enhancing visitors' understanding of agricultural heritage systems, thus raising tourist awareness for heritage conservation. Considering the high potential and the current low level of heritage interpretation, theoretical and managerial implications are then discussed to enhance agricultural heritage interpretations in both content and form to support the conservation and sustainable development of such dynamic agricultural heritage systems.

**Keywords:** heritage interpretation; agricultural heritage system; tourist perception; Qingtian rice-fish system

## 1. Introduction

As an important type of heritage, agricultural heritage is a living and evolving system of human communities in an intricate relationship with their territory, cultural or agricultural landscape, or biophysical and wider social environment [1,2]. The Globally Important Agricultural Heritage Systems (GIAHS) launched by the Food and Agriculture Organization of the United Nations (FAO) in 2002 [3,4] intends to promote the identification and conservation of such heritage and its associated landscapes, agricultural biodiversity, knowledge systems, and culture [3–5]. According to FAO, "a GIAHS is a living, evolving system of human communities in an intricate relationship with their territory, cultural or agricultural landscape or biophysical and wider social environment" [3]. As the GIAHS program continues to advance, the resilience, sustainability, and integrity of these important agricultural heritage systems are being emphasized and protected, which also contributes to the regional and global sustainable development [6,7].

Heritage tourism is regarded as a dynamic and important tool to support the conservation of agricultural heritage systems, as well as facilitate local economic and social development [5,8,9]. Agricultural heritage tourism could significantly enhance the educational and scientific values of heritage sites, which makes it different from rural tourism and agricultural tourism [10,11].

Over the last few decades, agricultural heritage tourism has attracted extensive research attention. Previous studies mainly focus on the conceptual exploration [2,6], tourism

development models [12–14], and community participation [3,10]. However, agricultural heritage tourism is still in the early stages of development with evident regional differences, which may result in low levels of community participation and incorrect heritage management approaches [4]. Therefore, enhancing the connection between tourists and agricultural heritage has become a significant research issue, which has enriched the academic understanding of this phenomenon.

Furthermore, heritage interpretation plays an essential role in strengthening the relationship between tourists and agricultural heritage sites [15–17]. Heritage interpretation in a tourism context refers to the educational activity that delivers and interprets information about natural and cultural heritages to people visiting heritage sites [16–18]. It not only enriches emotions and experiences between tourists and heritage resources, but also enhances awareness of heritage conservation and protection [18–21]. It is particularly true for agricultural heritage tourism, where dynamics and values of such human–nature systems need to be interpreted to tourists to achieve a better understanding of heritage values, stimulate interests, enhance tourism experiences, and develop an enhanced support for conservation [20,21]. However, little attention has been paid to the agricultural heritage interpretation, especially analysis from interpretation contents and forms. Therefore, the systematic construction of agricultural heritage interpretation system has become an essential research topic that would effectively improve heritage value recognition and enrich heritage tourism experiences.

To fill the above-mentioned research gaps, the research investigates the agricultural heritage interpretation through the gaze of tourists. Firstly, visitors' awareness of the interpretation contents in the rice-fish system is analyzed based on the interview and questionnaire methods. Then, tourist perceptions on the interpretation forms are explored according to the proposed framework of agricultural heritage interpretation (Figure 1). Our research makes two contributions. Theoretically, this study is among the first to explore agricultural heritage interpretation from both contents and forms. Practically, this study puts forward some managerial implications for improving the efficiency of agricultural heritage interpretation, which can provide more targeted and accurate references for policymakers to support further sustainable development.

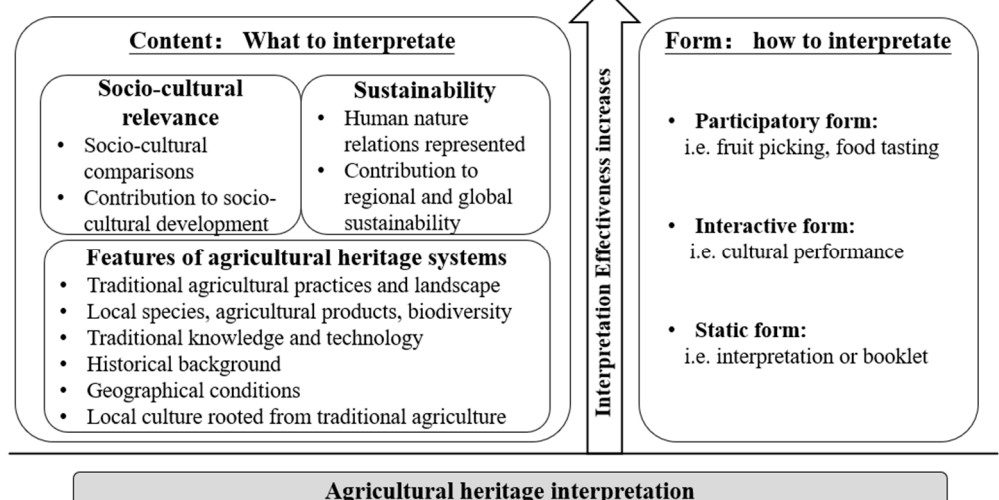

**Figure 1.** Two-dimensional framework of agricultural heritage interpretation.

The rest of this study is structured as follows. We reviewed related literature in Section 2. The materials and methods section presents the study area, data collection, and processing procedures in Section 3. Empirical results are reported in Section 4, in which we describe tourist perceptions on the contents and forms of agricultural heritage interpretation.

Section 5 presents final discussions and theoretical and managerial implications. Section 6 provides the conclusions of this work and future research directions.

## 2. The Literature Review

"Interpretation" was introduced as a professional term to refer to the "natural guidance" of natural adventure activities in the 1930s [21]. Studies in a wide range of settings indicate that the interpretation system plays important roles at heritage sites and would contribute to the ecological and socio-cultural sustainability at heritage destinations and enhance visitor experiences and emotional attachments [22–24]. Moreover, the development and popularization of digital technology have promoted the diversity of interpretation forms at tourist destinations [25,26]. Studies reveal that both what is interpreted and how it is interpreted would influence whether information is effectively disseminated and received [16–18,27,28]. Therefore, to enhance heritage interpretation effectiveness, it is necessary to look into both the content and forms of interpretation.

For agricultural heritage systems, a complex series of unique features need to be interpreted to tourists with a wide range of backgrounds through different forms of interpretation. Constantly evolving, agricultural heritage systems are composed of a complex mix of elements including traditional agricultural practices suited to local environment, local knowledge, skills, and technology that support such agricultural practices, local culture, and many cultural manifestations that are rooted from such practices, all of which are essential tourism resources with high interpretive value [1,2]. Moreover, as the creators and practitioners of agricultural heritage, local residents are critical components of agricultural heritage systems and well equipped to participate in heritage interpretations [18]. Thus, the contents of agricultural heritage interpretation would need to include the historical, social, and cultural background, geographical conditions, diverse features of agricultural practices, products, and traditions. Particularly, how such systems represent the intricate human–nature relations and contribute to regional and global sustainability are critical aspects that should be interpreted.

Therefore, presented by forms such as brochures, videos, and activities, agricultural heritage interpretation could help tourists understand and experience agricultural heritage, thus realizing the educational and research functions of agricultural heritage systems. Drawing from specific features of agricultural heritage systems and previous research on heritage interpretation, our research constructs a two-dimensional framework for agricultural heritage interpretation with interpretation contents and interpretation forms (Figure 1). In the two-dimensional framework, interpretation content refers to what should be interpreted to tourists, while interpretation form refers to ways and mediums to convey and disseminate such contents. Interpretation contents contain three aspects of socio-cultural relevance, sustainability, and features of agricultural heritage systems. While, interpretation forms include participatory form, interactive form and static form.

Forms of interpretation are usually categorized into static, interactive, and participatory. Static interpretation refers to static information provided to visitors through text materials, pictures, video, and other devices. Visitors can receive information from static-type interpretation passively, but there is no time limit on the access to such information. Interactive interpretation refers to active and dynamic ways to convey information to tourists, such as guided tours and cultural performances. Tourists could interact in the process and get tailor-made information. For participatory interpretation, tourists directly get involved in agricultural heritage practices and associated activities to learn and understand elements of such heritage, such as tasting local specialties, local food making, and participating in farming practices.

In combination, both content and form of interpretation would affect interpretation effectiveness. When interpretation contents get enriched from basic features to socio-cultural relevance and sustainability and forms of interpretation go from static to participatory, heritage interpretation would achieve higher effectiveness in ways of increasing tourists'

understanding of heritage, enhancing their recognition of heritage values, and prompting their support for conservation.

Engaging this two-dimensional framework as illustrated in Figure 1, this study aims to examine tourist response to agricultural heritage interpretations and identify strategies to enhance interpretation effectives at agricultural heritage sites. The Qingtian rice-fish system in Zhejiang Province was selected as the case study site with its relatively well-established tourism development among agricultural heritage sites in China.

## 3. Materials and Methods

### 3.1. Study Area

Qingtian County is located in the middle south of Zhejiang Province, China, and at the lower reaches of Ou River, which covers a total area of 2493 km$^2$. The rice-fish agricultural system has been practiced in the local area for more than 1000 years, which has become an important way of living and supports the unique local culture and traditions. There are about 475,153 people residing in the county, most of them are working abroad. In addition to agricultural income, local people mainly rely on salaries from abroad to support their lives.

Designated by FAO as a Globally Important Agricultural Heritage System (GIAHS) in 2005 among the first batch in China, the Qingtian rice-fish system (Figure 2), with its value highly recognized, generates substantial economic and social benefits for the local community through provisions of multiple goods and services for tourists. The number of tourists grew from 0.67 million in 2005 to 7 million in 2018, and tourism revenue increased from 0.74 billion yuan in 2005 to 14.05 billion yuan in 2018 (as shown in Figure 3). In 2015, the government invested in the construction of a 5A tourist attraction in Shimen Cave and developed an agricultural heritage tour with Fangshan Township as the core. However, facing drastic changes induced by tourism, the construction and development of an interpretation system, an important component of agricultural heritage tourism, has not received much attention in the rice-fish system.

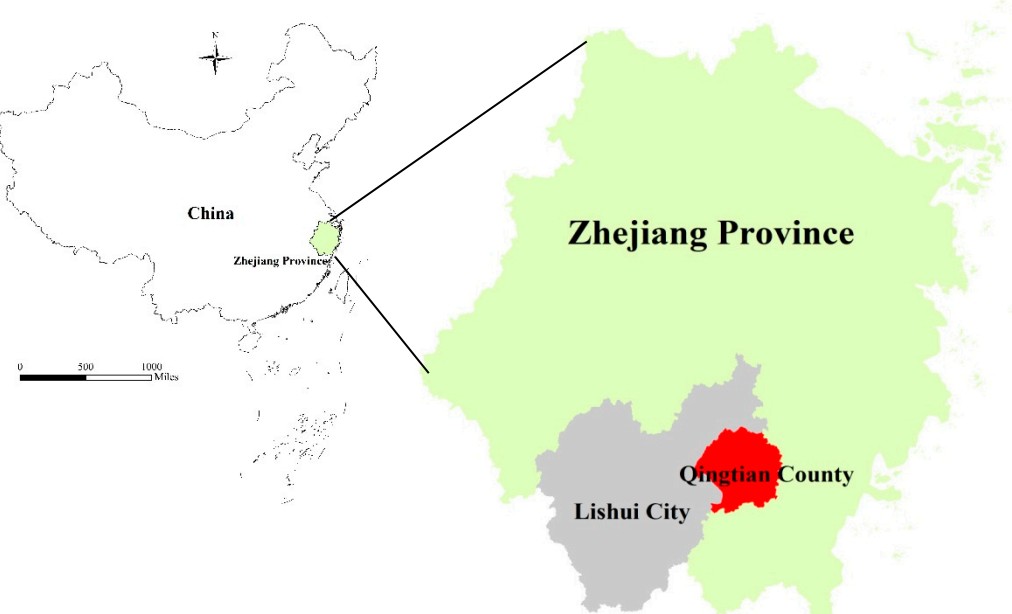

**Figure 2.** Location of rice-fish system, Qingtian County, Zhejiang Province.

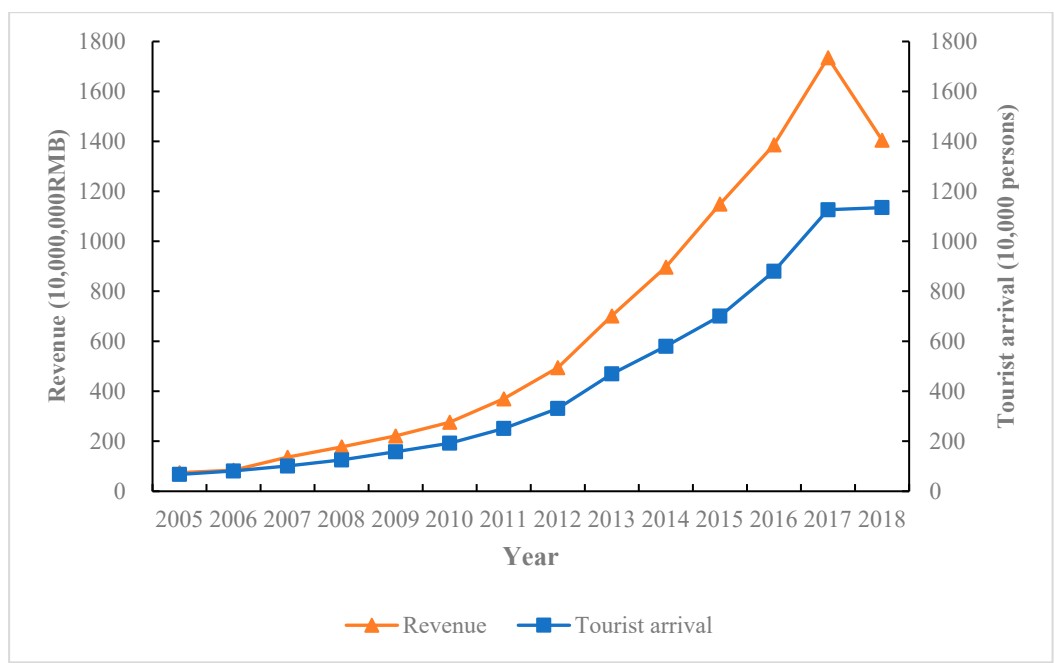

**Figure 3.** The growth of tourist arrival and tourism revenue in Qingtian County.

Longxian village is a typical heritage village in the core protection area of the Qingtian rice-fish system, covering an area of 4.6 km² (Figure 4). Although there are 510 registered residents, less than half currently reside in the village. According to the local statistics, over 650 people of Longxian village are still living abroad. Many overseas relatives supply local residents abundant market information and spread the village's products to the world. Every year hundreds of overseas relatives, tourists, and merchants visit the village and the village's fishing products, tea, carved stone, and other products are taken to their countries of residence. In recent years, the tourism of Longxian village has developed rapidly. During the survey period, there were five restaurants and one hotel in the village. The restaurants and hotel have good facilities with high service quality.

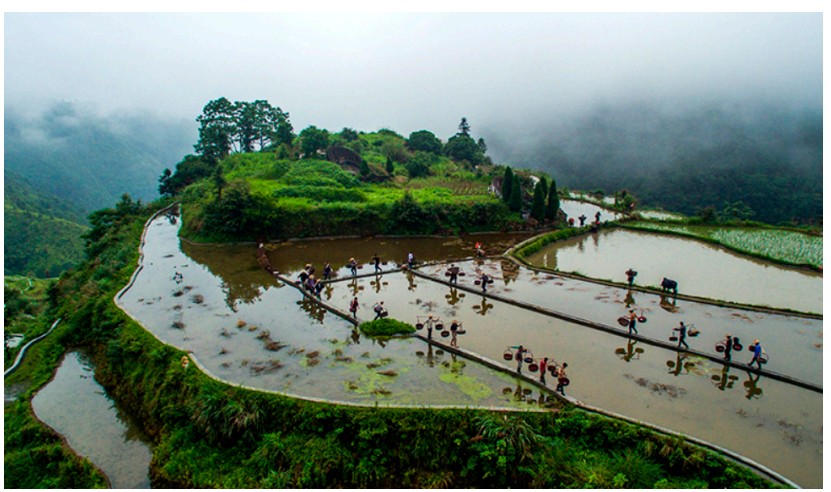
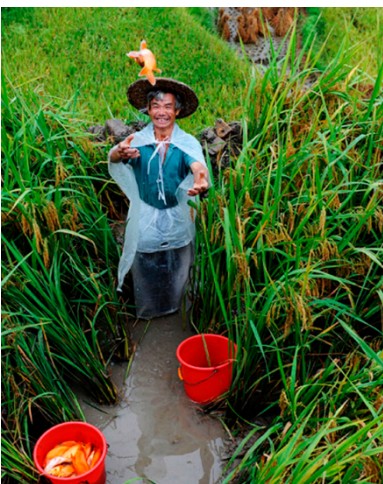

**Figure 4.** The panoramic view of the rice-fish system (photos provided by National Agricultural Exhibition Center).

### 3.2. Data Collection and Processing

In addition to field investigations and GIS analysis for resource mapping, face-to-face semi-structured interviews and a questionnaire survey were chosen as primary data

collection methods to evaluate the tourists' perceptions of heritage interpretation. Data from primary and secondary sources were collected during two field surveys for a total of 53 days from 29 July to 20 August 2018 and 15 July to 12 August 2019. Observations were conducted during the two field investigations to understand tourists' perspectives on agricultural heritage interpretations. Field notes were taken to document what was observed and experienced.

During the first phase, face-to-face in-depth interviews were conducted from July 29 to 20 August in 2018. Twenty-three tourists were obtained using a convenience sampling method. The interview process stopped when no new information could be obtained. Interviews intended to understand tourist perception and evaluation of interpretations of the rice-fish system, which were carried out through a dialogue form to assure their understanding of interview questions and information obtained. The total interview time was 305 min. Seven interviews with interview time less than 5 min were excluded, and the average interview time per tourist for the remaining 16 valid interviews was 18 min.

Based on the analysis of field interviews, the visitor questionnaire survey was conducted from 15 July to 12 August in 2019. The questionnaire was designed with three parts. Part 1 contained visitors' basic trip information to the rice-fish system. Part 2 applied five-point Likert scale questions to examine the tourists' motivations and perceptions. Part 3 collected the demographic and socio-economic characteristics. Prior to data collection, investigators were briefed about the overall research purpose, the research procedure, and basic survey skills. A total of 247 questionnaires were distributed, with 230 valid questionnaires obtained. Only data related specifically to the research content was presented in this paper.

The interview texts were analyzed by ROST CM6 software to classify and analyze key words related to interpretation resources. Quantitative data from the questionnaires were analyzed by SPSS 20.0 to conduct descriptive statistical analysis of the questionnaires.

## 4. Results

### 4.1. Characteristics of Respondents

The basic information of questionnaire sample can be seen in Table 1. The gender ratio of tourists was basically balanced and the age of tourists was mainly 21–50 years old (87.8%). Many visitors were well educated (70.4% with high school or above), with self-employed (24.4%) as the major career cohort. In addition, farmers composed an important group of tourists with a share of 11.30%. Finally, 37.0% of the tourists had a monthly income of 3001–6000 RMB.

**Table 1.** Basic information of questionnaire respondents.

|  |  | **Number** | **Proportion** |
|---|---|---|---|
| Gender | Male | 117 | 50.9% |
|  | Female | 113 | 49.1% |
| Age group | 20 and under | 12 | 5.2% |
|  | 21–30 | 64 | 27.8% |
|  | 31–40 | 84 | 36.5% |
|  | 41–50 | 54 | 23.5% |
|  | 51–60 | 12 | 5.2% |
|  | 61 and above | 4 | 1.7% |
| Education | Junior high school or less | 68 | 29.6% |
|  | Senior high school | 72 | 31.3% |
|  | College | 65 | 28.3% |
|  | Master and above | 25 | 10.9% |

**Table 1.** *Cont.*

| | | Number | Proportion |
|---|---|---|---|
| | Government employee | 27 | 11.7% |
| | Public institution | 30 | 13.9% |
| | Self-employed | 56 | 24.4% |
| Career | Company employee | 43 | 18.7% |
| | Student | 40 | 17.4% |
| | Farmer | 26 | 11.3% |
| | Others | 8 | 3.5% |
| | Less than 3000 RMB | 66 | 28.7% |
| | 3001–6000 RMB | 85 | 37.0% |
| Income | 6001–9000 RMB | 47 | 20.4% |
| | 9001–12,000 RMB | 18 | 7.8% |
| | Higher than 12,000 RMB | 14 | 6.1% |

Table 2 demonstrates the basic information of the interview sample. Face-to-face in-depth interviews were conducted with 16 visitors, 9 of whom were male and 7 female. The age of tourists was mainly 31–60 years old. In comparison, the interview sample showed lower education (68.8% with junior high school or below) and higher income (50.0% with 6000–12,000 RMB). Self-employed was the major occupation (37.5%), which is consistent with the questionnaire sample.

**Table 2.** Basic information of interviewees.

| | | Number of Interviewees |
|---|---|---|
| Gender | Male | 9 |
| | Female | 7 |
| | 20 and under | 0 |
| | 21–30 | 2 |
| Age group | 31–40 | 5 |
| | 41–50 | 4 |
| | 51–60 | 4 |
| | 61 and above | 1 |
| | Junior high school or less | 11 |
| Education | Senior high school | 3 |
| | College | 1 |
| | Master and above | 0 |
| | Government employee | 1 |
| | Public institution | 3 |
| | Self-employed | 6 |
| Career | Company employee | 1 |
| | Student | 0 |
| | Farmer | 1 |
| | Others | 5 |
| | Less than 3000 RMB | 3 |
| | 3001–6000 RMB | 2 |
| Income | 6001–9000 RMB | 4 |
| | 9001–12,000 RMB | 4 |
| | Higher than 12,000 RMB | 2 |

*4.2. Agricultural Heritage Resources Mapping*

The long history of the rice-fish system has led to a rich tradition of rice-fish culture, not only in local knowledge and tools for agricultural practices, but also in local customs, festivals, cuisine, and so on. Through field investigations and secondary data collection, mapping of agricultural heritage resources in Longxian village was conducted. As shown

in Table 3, 1008 agricultural heritage resource points were identified in Longxian village. With wide varieties of agricultural products including cowpeas, beans, cucurbits, and leeks, numerous local vegetable species are identified as the most abundant agricultural heritage resources. Moveover, cultural resources include not only tangible traditional village residences and other functional buildings, but also intangible traditional art and cultural performances rooted with the rice-fish systems.

**Table 3.** Agricultural heritage resources of the rice-fish system.

| Type of Contents | Specific Types | Specific Expressions | Points |
|---|---|---|---|
| Traditional knowledge and technology | Rice-fish field | Rice and fish integrated farming | 74 |
| | Rice field | Only rice without fish | 3 |
| Biodiversity | Vegetables | Cowpeas, beans, cucurbits, kidney beans, string beans, cucumbers, loofahs, leeks, eggplants, peppers, okra, amaranth, bitter gourd, bitter greens, bitter greens (wild), winter squash, eight-sided gourd, knife beans, pumpkins, yams, root beans, broomrape, hairy taro, ginger, taro, ginger, bitter flax, bitter dice, cabbage, radish, mustard greens | 700 |
| | Fruit trees | Chestnuts, prunes, peach trees, persimmons | 27 |
| | Crops other than rice | Cotton, peanuts, sweet potatoes, corn, black beans, green beans | 145 |
| | Chinese medicinal materials | Fishy herb, iron horse whip | 50 |
| Cultural elements | Traditional architecture | Wu Qiankui's former residence, Wu's ancestral hall | 2 |
| | Modern buildings | Niangniang temple, Church | 3 |
| Natural resources | Natural resources | Qiyun Mountain, Eighteen Pools | 2 |

The distribution of tangible agricultural heritage resources at Longxian village illustrates two clusters, one is near Wu Qiankui's former residence in the southwest of the village and the other is near the church in the middle of the village. Clustered with elements of traditional agriculture, human landscape such as Wu's ancestral hall, Niangniang temple, and church, and natural landscape such as Qiyun Mountain, the village center is also equipped with service facilities such as restaurants to support tourism development.

*4.3. Tourist Perceptions on the Contents of Current Heritage Interpretation*

Text mining of the interview texts was performed and the most common words in the interview texts, consisting of at least two letters, were identified [22]. The word cloud was generated using ROST CM6 software. A total of 33 high-frequency words were obtained after removing quantifiers, meaningless transitions and verbs, and words not related to heritage interpretation during the screening of the interview texts. The words were divided into five categories of heritage interpretation contents, including historical background, geographical conditions, natural resources, cultural elements, and traditional agriculture (Figure 5).

The word cloud analysis of interview transcripts shows that features of traditional agriculture, in particular the rice-fish system, agricultural products, and food experiences, were highly recognized among current heritage interpretation contents at Qingtian. Tourists demonstrated their interests and high evaluation of the rice-fish field experiences and local food tasting, such as Shanfen dumplings and Tanggao. However, tourist recognition of local cultural and natural resources was relatively low.

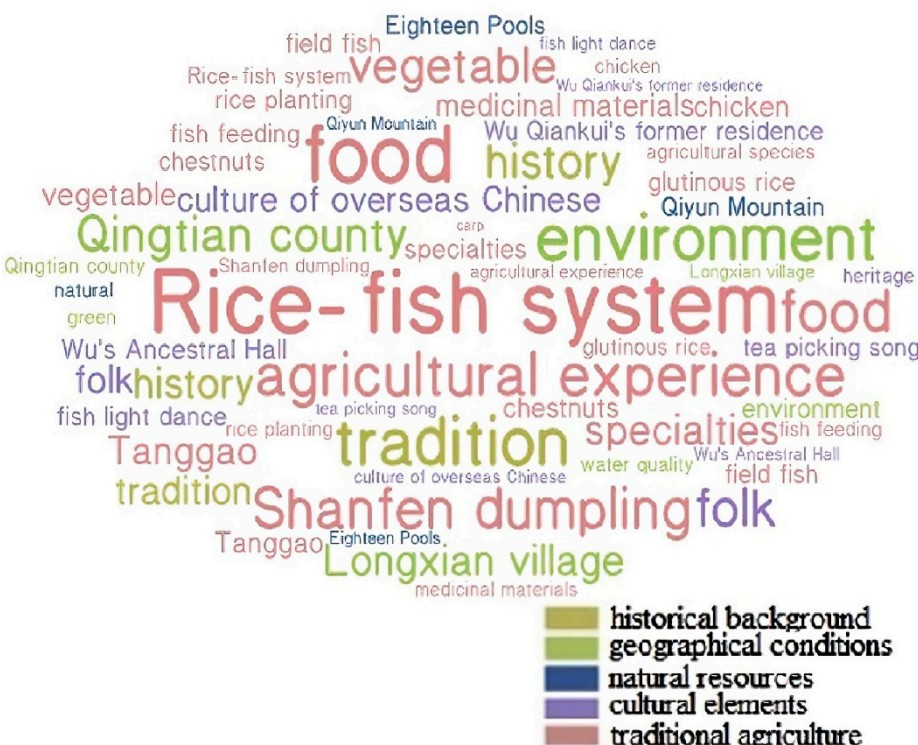

**Figure 5.** The word cloud of tourist perception on the interpretation contents.

Semantic network analysis is then applied to analyze the relevance and hierarchy of the interpretation content as perceived by tourists. Figure 6 shows that "rice-fish system" is the first-level vocabulary, and "agricultural experience", "food", and "environment" are the second-level terms. Among them, "rice-fish system" is associated with "Qingtian County", "Longxian Village", "heritage", and "tourism". The term "food" includes "Shanfen dumpling", "Tanggao", "field fish", and "vegetables". It can be seen that tourists were more inclined to taste local food. However, the low centrality of the semantic network diagram reflects that tourist perceptions on the contents of agricultural heritage interpretation were relatively scattered.

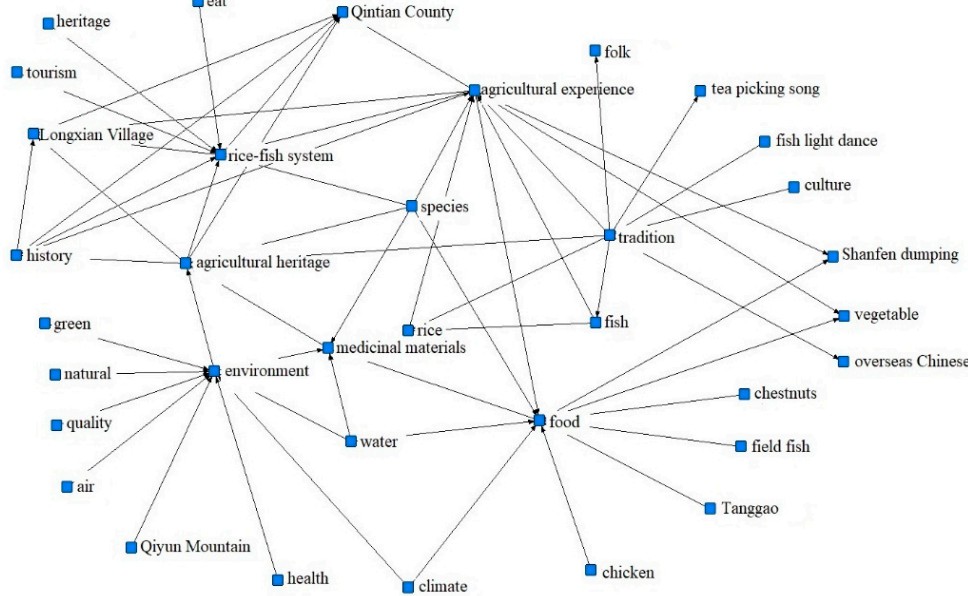

**Figure 6.** The semantic network of high frequency words.

To further quantify tourists' recognition of agricultural heritage interpretation, descriptive statistical analysis of the questionnaire data was performed using SPSS 20.0. The mean value represents the overall tourist perception on each indicator, while the standard deviation indicates the degree of dispersion. As shown in Table 4, the degree of tourist recognition on agricultural heritage was generally low despite rich heritage interpretation provisions at Longxian village. Only the means of recognition of local vegetables and rice-fish fields were over 3, indicating moderate positive recognitions. However, tourist recognition for traditional culture, knowledge, and technology were low, with means of all items between 2 and 3, indicating a moderate negative recognition. In addition, the standard deviation was higher than 1, which indicates that there were obvious differences in tourist perceptions of interpretation contents. Such results indicate a currently low level of tourist recognition of features of agricultural heritage systems. However, primarily static interpretations at museums, though well developed and covering a breadth of the interpretation contents, do not fully arouse visitors' interests in learning about agricultural heritage systems.

**Table 4.** Tourist recognition of components of the rice-fish system.

| Types | Specific Types | Number | Mean | Standard Deviation |
|---|---|---|---|---|
| Traditional agriculture landscape | Rice-fish field | 230 | 3.155 | 1.521 |
| | Rice field | 230 | 3.592 | 1.356 |
| Biodiversity | Vegetable | 230 | 3.646 | 1.347 |
| | Wild Vegetable | 230 | 3.223 | 1.399 |
| | Chinese traditional drug | 230 | 2.631 | 1.415 |
| | Wild herbal medicine | 230 | 2.323 | 1.272 |
| Traditional knowledge and technology | Rice seedling knowledge | 230 | 2.685 | 1.549 |
| | Rice cultivation technology | 230 | 2.623 | 1.405 |
| | Traditional fish hatchery | 230 | 2.208 | 1.196 |
| | Agricultural tools | 230 | 2.977 | 1.454 |
| Traditional culture | Fish light dance | 230 | 2.885 | 1.542 |
| | Folk activities | 230 | 2.531 | 1.426 |

Note: 5-point Likert scale is used to measure tourist recognition with 1 indicates the lowest recognition and 5 indicate the highest recognition.

### 4.4. Tourist Perceptions on the Forms of Agricultural Heritage Interpretation

Possessing rich agricultural heritage resources and being the first and probably most visited GIAHS sites in China, agricultural heritage interpretation has been well established in Qingtian and is currently in the development stage with efforts from the local government, the village, and local residents and relatives from abroad. Through field research and review of relevant documents and websites, the current provision of heritage interpretation of Qingtian rice-fish system in Longxian village was mapped and categorized according to the proposed framework of agricultural heritage interpretation (Figure 1 and Table 5).

Examples of static, interactive, and participatory forms of agricultural heritage interpretation at Qingtian are illustrated in Figure 6. First, in addition to interpretation boards and demonstrations in the field, a series of museums have been established mainly as the major form of the static heritage interpretation (Table 5). On top of providing a comprehensive demonstration of features, functions, and values of the rice-fish system, current museum development at Qingtian also illustrates the importance of agricultural heritage systems to regional and global sustainability and tries to excavate cultural elements relevant to tourists of different backgrounds, such as red culture, honest culture, and culture and traditions of overseas Chinese.

**Table 5.** Existing forms of interpretation provision on Qingtian rice-fish systems.

| Interpretation Form | Existing Interpretation Provision in Qingtian | Interpretation Content |
|---|---|---|
| Static | In the field display of GIAHS information and key elements of Qingtian rice-fish system<br>A series of museums have been established to interpret different features and values of the Qingtian rice-fish system to visitors:<br>● 4 Museums of agricultural science and technology<br>● 2 Museums of traditional agriculture practices<br>● 3 Museums of history of overseas Chinese<br>● 3 Museums of Nostalgia: traditions in old days<br>● 1 Museum of red culture<br>● 1 Museum of local honest culture | Features of agricultural heritage<br><br>Features of agricultural heritage touching on sustainability<br><br>Socio-cultural relevance to visitors from various backgrounds |
| Interactive | Performance of Qingtian fish light dance, tea picking song, and so on | Features of agricultural heritage + socio-cultural relevance |
| Participatory | Taste local food and participate in food making (i.e., dumplings, toufu)<br>Participate in agricultural practices (i.e., rice transplanting, fishing feeding) | Features of agricultural heritage + socio-cultural relevance |

Interactive interpretations are mainly provided as cultural performances with local features. In particular, the Qingtian fish light dance (Figure 7, lower left) is recognized as an important component of the rice-fish agricultural system, which also serves as an interactive activity to enrich tourist understanding of the human–nature relations supported by the rice-fish system at Longxian Village in Qingtian County. However, Qingtian fish light dances demonstrated by local residents are usually held from January to February, while the tourism peak season is from May to August every year. Therefore, it is difficult for tourists to experience such traditional cultural activities. As one tourist put forward:

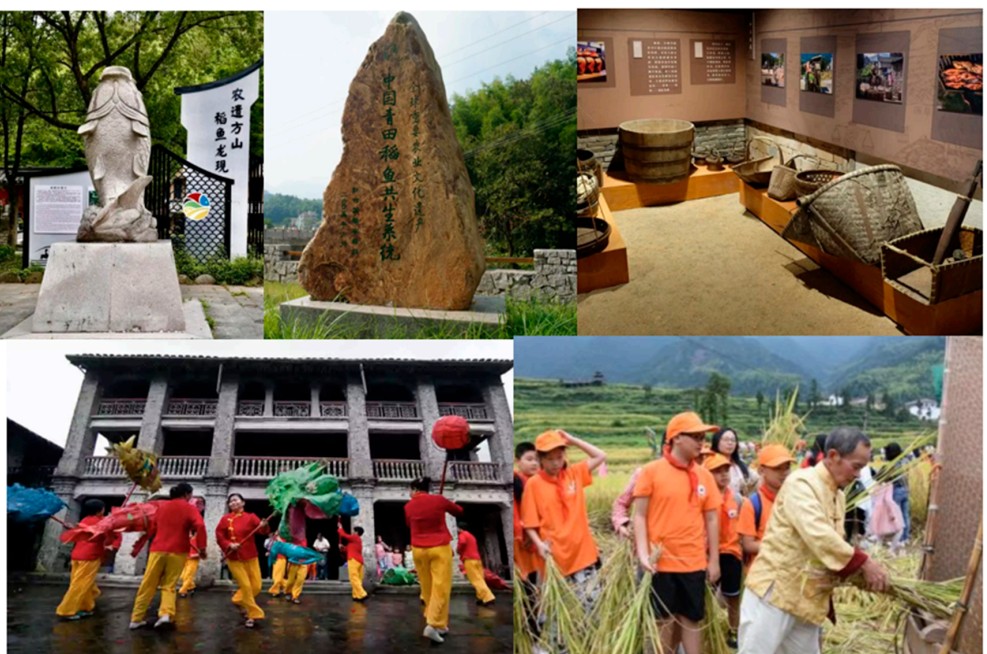

**Figure 7.** Demonstrations in the field (**top left**), Qingtian agricultural heritage exhibition center (**top right**), Qingtian fish light dance (**down left**), and the educational tour of the rice-fish system (**down right**).

*"We have heard that Longxian Village has many traditional cultural activities. However, we did not experience these cultural activities, such as the Qingtian fish light dance. The food here is very attractive to us and we also enjoy it."*

The rice-fish system consists of many experiences and technologies that tourists could learn through direct participation. Educational tours and participation in agricultural practices are gaining popularity among service providers and tourists in recent years as the major forms of participatory interpretations. In addition to agricultural practices in the rice-fish fields, participatory interpretations are getting enriched through tourist participation in traditional cultural activities, food making, and tasting.

Moreover, interviews and field observations show tourists' high levels of interest in experiencing local delicacies such as rice fish, Shanfen dumplings, and Tanggao. By participation in the making and tasting of such local food, tourists not only acquire a positive tourist experience by enjoying local food culture, but also further extend their understanding of the value of the Qingtian rice-fish system, trigger tourist spending, and enhance tourist attachment to heritage sites. As one tourist shared:

*"In the past, we know only there are rice-crab fields in the north east of China, today we learned the rice-fish system in Qingtian. The view of red fish swimming in the green rice fields is beautiful and soothing . . . The best part of our trip is eating fish we caught in the fields ourselves and the taste is very fresh even with the simplest way of cooking . . . We also brought some dried fish and learned the cooking methods from our host."*

## 5. Discussion and Implications

### 5.1. General Discussions

Integrating both qualitative interview and quantitative questionnaire, research results reveal that, as an important tool to enhance visitors' heritage experiences, create emotional attachments to heritage resources, and raise public awareness of heritage values and willingness to support its conservation [16,20,25], a well-developed heritage interpretation system is critical for agricultural heritage conservation.

Based on the case of the rice-fish system, current provision of agricultural heritage interpretation was mapped. Then, visitor interviews and questionnaires were engaged to examine the effects of interpretation contents and interpretation forms on tourists' heritage recognition and experience evaluation of Qingtian rice-fish systems. The research findings reinforce the importance of both contents and forms of heritage interpretation in sharping tourists' recognition and evaluation of agricultural heritage, which echoes with research of other agricultural heritage sites [16,20,25]. Different from other types of heritage [6,16,25], agricultural heritage systems are rich with components that could be developed for participatory interpretations, such as participating in agricultural practices, agricultural product tasting, learning the traditional culture, and many more. Thus, interpretation for agricultural heritage systems has the potential to be further enriched and diversified both in contents and in forms. In addition, the two-dimensional framework on agricultural heritage interpretation of contents and forms has been proven to be a useful framework to understand and evaluate interpretation effectiveness.

Research shows that current provisions of heritage interpretation has covered well-rounded content; in particular, dozens of museums of different focuses provide rich static learning opportunities for tourists and residents. However, such static heritage interpretation is not very effective in arousing tourist interests of learning. Current recognition of tourists is primarily based on basic and superficial features of the rice-fish system, such as agricultural landscape and local products. Such results are similar to studies of other GIAHS sites such as the Duotian Agrosystem and Xuanhua Grape Garden [2,10]. In comparison, contents of interactive and participatory forms of interpretations are better received by tourists. The current low and somewhat superficial tourist recognition of Qingtian rice-fish systems indicates that current interpretation provision still needs to be improved. In addition to the need to enrich and deepen interpretation contents [25], diverse forms of heritage interpretation need to be engaged, in particular interactive and participatory

forms, to enhance tourist interactions and participations in leaning, experiencing, and understanding values of agricultural heritage. In particular, traditional culture, knowledge, and technology rooted from agricultural practices in rice-fish fields which are not highly recognized by tourists at the current stage as shown in this research, have high potential to enrich current heritage interpretation both in content and form.

### 5.2. Theoretical and Managerial Implications

The current research has some substantial theoretical implications. Firstly, responding to limited research on the effectiveness of agricultural heritage interpretation, this study is among the first research efforts to build the two-fold framework for agricultural heritage interpretation, which can more systematically and objectively reflect the interpretation. This is conducive to enriching the research content regarding the interpretation system. This issue concerns not only behavior in areas of sustainable development but in general the validity of interpretation. Secondly, previous studies mostly focused on components of interpretation, including interpretation objects, interpretation techniques, interpretation methods, and interpretation effects [14,16–18,22,27]. With the Qingtian rice-fish system as an example, this study complements the literature on heritage interpretation by analyzing tourist perceptions of heritage interpretation from contents and forms.

The development of effective heritage interpretation is crucial to the sustainable development of agricultural heritage sites and communities. Managerial implications to enhance heritage interpretation and improve effectives in information and value dissemination to visitors are then generated for the Qingtian rice-fish system.

First, in terms of interpretation forms, interactive and participatory interpretations demonstrate high effectiveness in enriching tourists' experiences and stimulating their interests in learning about heritage, while static interpretations have certain limitations. Although there are many traditional cultural activities in Longxian village, such as Qingtian fish light dance and tea picking song, many tourists have difficulty experiencing the traditions due to mismatch of seasons. Therefore, the interpretation system could be designed with the help of new technologies such as artificial intelligence and multimedia channels to extend the spatial and temporal dimensions of heritage resources and extend the availability of interpretation, so that tourists could experience and participate with a wider flexibility.

Second, with the changing status of tourists and evolving technology, how tourists receive information and their preferences for information are dynamically evolving. Therefore, the design of the interpretation system needs to respond to such dynamic and segmented market demand. Development of static heritage interpretation should reflect specific needs and interests of different groups of tourists, such as story illustration books for children, study activities with different agricultural themes for students, and interpretation of health functions for elder visitors. In addition, multimedia technology should be integrated into the agricultural heritage interpretation system in order to provide optimal experience and emotion to visitors. The variety of interpretation experiences will allow tourists from different age groups to better understand, appreciate, and enjoy experiences with agricultural heritage systems.

Finally, community plays a vital role in GIAHS conservation and tourism development. Innovative measures should be taken to further enrich the means of community participation in heritage conservation and to strengthen the local identity awareness of community. Local authorities or administrators of agricultural heritage sites should build a professional team of community interpreters to tailor-make and personalize heritage interpretations through interactive and personalized ways of communications, which would ease the way of information acquisition and enhance visitor's emotional attachments to heritage resources and the heritage community.

## 6. Conclusions

To reflect the need to develop effective heritage interpretations to enrich heritage tourism experiences and enhance public awareness of heritage values, this study aims to explore how tourists perceive and evaluate heritage interpretations with the proposed framework of heritage interpretation contents and forms through the case study of the rice-fish system at Qingtian, Zhejiang Province of China. Key research questions include what the status of current heritage interpretation provisions is and how and to what extent tourists perceive and respond to different interpretation contents and forms of interpretation.

Research identified the abundant agricultural heritage resources of the rice-fish system that have the potential to be used for tourism. However, field investigations and face-to-face in-depth interviews identified that tourists' overall recognition of agricultural heritage features of Qingtian rice-fish system was generally low. From the perspective of interpretation contents, visitors' awareness of the rice-fish agricultural landscape, traditional cuisine, and biodiversity was relatively high. In terms of forms of interpretation, consistent with previous studies in the Duotian Agrosystem, Jiangsu Province of China [2], compared to the static and interactive interpretations, participatory interpretations demonstrated higher effectiveness to enrich tourists' knowledge and understanding of the agricultural heritage system. Then, theoretical and practical implications were discussed to support the construction of an effective agricultural heritage interpretation system that could pass on the rich cultural and natural heritage resources to tourists in contents and forms that are welcomed by them.

In this study, tourists' perceptions of agricultural heritage interpretation are investigated based on an empirical analysis. However, this research has limitations to be addressed in the future. First, restricted by time and resources, only a limited number of tourists were approached and successfully interviewed. A wider range of visitors at Longxian village, Qingtian county could be reached out to in order to improve the representativeness in future research. Second, the interpretation framework of agricultural heritage proposed and tested in this study has proven its usability in understanding the effectiveness of interpretation, which could be further applied and refined based on field research at other agricultural heritage sites, thus promoting the sustainability of agricultural heritage systems. Moreover, comparative research can be conducted at different agricultural heritage sites in China and other regions and countries to understand needs for interpretation in different cultural and geographical contexts.

**Author Contributions:** M.W. and M.S. drafted the manuscript, Y.W. conducted the field research and carried out part of the research, Y.S. and M.S. developed the research framework, Y.S. supervised the field investigation. All authors have read and agreed to the published version of the manuscript.

**Funding:** This work was supported by the National Natural Science foundation of China (project number 41971264 and 42171232).

**Institutional Review Board Statement:** Not applicable.

**Informed Consent Statement:** Informed consent was obtained from all subjects involved in the study.

**Data Availability Statement:** The data are available from the corresponding author upon reasonable request.

**Acknowledgments:** Sincere thanks to all respondents for their participation in the research.

**Conflicts of Interest:** The authors declare no conflict of interest.

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
