# Peer review of "Tourist Perspectives on Agricultural Heritage Interpretation—A Case Study of the Qingtian Rice-Fish System"

_sustainability, doi:10.3390/su141610206_

Round 1
Reviewer 1 Report
The research and the topic are interesting, but some essential parts of the article are missing. The article also needs to be reorganized.
Introduction - please emphasize why chosen topis is important.
The Literature Review section is missing. In this section, please introduce methods that were used for the examination of agricultural heritage interpretation. Please emphasize the definition.
Figure 1 should be the result of the literature review with methods. Move it from the Introduction.
After the Literature review should be the Methods section and then the Results section
Mapping should be part of the result section.
Please add the demographic characteristics. How was the research sample selected? Is it representative?
Results should contain an answer to the research question and achieve the aim of the study. In conclusion, in connection to the aim of the study, you should add Theoretical and Managerial implications
Author Response
Dear reviewer:
Thank you for your letter accepting the manuscript entitled Tourist Perspectives on Agricultural Heritage Interpretation – A Case study of Qingtian Rice-fish System. We were pleased to know that our work was rated as potentially acceptable for publication in Sustainability. We thank the reviewers for the time and efforts that they have put into reviewing the original manuscript. We have made changes according to comments of two reviewers. Accordingly, we have uploaded the revised version of the article with all changes highlighted in red.
- Response to comments by Reviewer #1
Main points:
- The research and the topic are interesting, but some essential parts of the article are missing. The article also needs to be reorganized. Introduction - please emphasize why chosen topis is important.
Response: Thank you for raising the concerns. According to the expert's suggestion, we have emphasized why the chosen topic is important and added the purpose of this research in introduction (Line 62~79 of Page 2).
- The Literature Review section is missing. In this section, please introduce methods that were used for the examination of agricultural heritage interpretation. Please emphasize the definition. Figure 1 should be the result of the literature review with methods. Move it from the Introduction. After the Literature review should be the Methods section and then the Results section.
Response: Thank you very much for this good suggestion. Following your suggestion, we have added and modified the Literature Review section. In the Literature Review, we have introduced methods that were used for examination of agricultural heritage interpretation and emphasized the definition (Line 105~113 of Page 3). Figure 1 has been moved out of the Introduction and placed in the Literature Review section (Line 124~126 of Page 3). Moreover, we have modified the structure of this paper. Section 1 presents the Introduction, Section 2 demonstrates the Literature review, Section 3 reviews the methodology and materials, Section 4 shows empirical results, Section 5 discusses the findings and theoretical and managerial implications, Section 6 draws conclusions and summarizes the shortcomings of the research.
- Mapping should be part of the result section.
Response: Thank you very much for raising the issue. Figure 2 shows the location of Rice-fish system, Qingtian County, Zhejiang Province. The section on agricultural heritage resources mapping is put in the second part of the finding section following the description of respondent characteristics.
- Please add the demographic characteristics. How was the research sample selected? Is it representative?
Response: Thank you very much for raising the issue. Following your advice, we have added the demographic characteristics in Table 1 and Table 2. Furthermore, we have the detailed description of the research sample selection process (Line 181~228 of Page 5~7).
- Results should contain an answer to the research question and achieve the aim of the study. In conclusion, in connection to the aim of the study, you should add Theoretical and Managerial implications.
Response: Thank you very much for this good suggestion. Following your advice, we have revised and improved the empirical analysis part and discussion part. Meanwhile, we have added theoretical and practical contributions to the discussion section to highlight the theoretical and managerial implications of the manuscript (Line 371~411 of Page 13).
We appreciate Editors/reviewers’ comments, and hope that the revision will be considered satisfactory. Once again, thank you very much for your comments and suggestions.
Yours sincerely,
All author
Reviewer 2 Report
1. Introduction
You should explain what agricultural heritage system is.
2. Materials and Methods
2.2. Data Collection and Processing
Sample description is usually part of Results chapter.
You need to provide more information about you sample design and data gathering process (both).
You interviewed 16 people, so calculating percentage is useless.
You need to describe data processing.
Did you use qualitative methods (interview) to get more information that was later used in quantitative research?
3. Results
Result are difficult to understand. I suggest that you add questions in your introduction (in relation to paper’s goal) and then structure this part accordingly. You could also separate your results based on the data collection method.
Author Response
Dear reviewer:
Thank you for your letter accepting the manuscript entitled Tourist Perspectives on Agricultural Heritage Interpretation – A Case study of Qingtian Rice-fish System. We were pleased to know that our work was rated as potentially acceptable for publication in Sustainability. We thank the reviewers for the time and efforts that they have put into reviewing the original manuscript. We have made changes according to comments of two reviewers. Accordingly, we have uploaded the revised version of the article with all changes highlighted in red.
- Response to comments by Reviewer #2
Main points:
- Introduction
You should explain what agricultural heritage system is.
Response: Thank you for raising the concerns. We have added the definition of Globally Important Agricultural Heritage System by FAO (Line 33~36 of Page 1).
- Materials and Methods
2.2. Data Collection and Processing
Sample description is usually part of Results chapter. You need to provide more information about you sample design and data gathering process (both). You interviewed 16 people, so calculating percentage is useless. You need to describe data processing. Did you use qualitative methods (interview) to get more information that was later used in quantitative research?
Response: Thank you very much for this good suggestion. Following your suggestion, we have moved the sample description to the Results section and revised the basic information of respondents in Table 1 and Table 2 (Line 213~228 of Page 6~7). We have provided more information about the sample design and data gathering process (Line 181~211 of Page 5~6). What’s more, qualitative methods (interview) were chosen as primary data collection methods to explore the tourists’ perceptions of heritage interpretation.
- Results
Results are difficult to understand. I suggest that you add questions in your introduction (in relation to paper’s goal) and then structure this part accordingly. You could also separate your results based on the data collection method. The status of agricultural heritage interpretation currently at the destination? To what extent tourists understand and evaluate the content and form of agricultural heritage interpretation?
Response: Thank you very much for raising the issue. Following your comment, we have added the research questions and purposes in the Introduction (Line 62~73 of Page 2). We divided the Results section into four subsections according to the two-dimensional framework of agricultural heritage interpretation system: 4.1 Characteristics of respondents; 4.2 Agricultural heritage resources mapping; 4.3. Tourist perceptions on the contents of current heritage interpretation and 4.4 Tourist perceptions on the forms of agricultural heritage interpretation (Line 212~339 of Page 6~12). We have added the status of agricultural heritage interpretation currently in Rice-fish system (Line 155~159 of Page 4; Line 274~276 of Page 9). We have elaborated tourists' perceptions on agricultural heritage interpretation using interview and questionnaire methods in 4.3 Interpretation contents and 4.4 Interpretation forms (Line 253~339 of Page 9~12).
We appreciate Editors/reviewers’ comments, and hope that the revision will be considered satisfactory. Once again, thank you very much for your comments and suggestions.
Yours sincerely,
All author

Round 2
Reviewer 1 Report
1. Please emphasize the aim of the study and research questions. Ansver on it in conclussions.
2. Qualitative results, please add more results and its analysis.
3. Discussion, what is similar and what is different from previos results of studies
Author Response
Response to Review
Dear Reviewer,
Thank you very much for your constructive comments and we have carefully revised our manuscript accordingly. All revised parts are highlighted in blue for your easy reference. The following is the point-to-point responses to your comments:
- Please emphasize the aim of the study and research questions. Answer on it in conclusions.
Response: We reiterated the aim of the study and our research questions in the first two paragraphs of section 6 Conclusions.
- Qualitative results, please add more results and its analysis.
Response: Thank you for raising the concerns. According to your suggestion, we added the semantic network of high frequency words qualitative results and analysis in section 4.3. Tourist perceptions on the contents of current heritage interpretation and more content analysis and quotations from interviewees in section 4.4 Tourist perceptions on the forms of agricultural heritage interpretation.
- Discussion, what is similar and what is different from previous results of studies
Response: Thank you for your insightful suggestions. We added comparisons of results of our study with previous studies in section 5.1 General discussions and section 6 conclusions. Both similarities and differences are discussed. We hope that in this way, our arguments can be strengthened.
Thank you again for your help to our manuscript again and we hope the revised version can be considered acceptable. Should you have any questions, please feel free to contact us.
Best regards,
All authors
